# VOC Emission from Lightweight Wood Fiber Insulation Board

**Dorota Fuczek [1,*], Magdalena Czajka [2], Jarosław Szuta [1], Krystian Szutkowski [1] and Patrycja Kwaśniewska-Sip [2]**

[1] Steico Sp.z.o.o., 64-700 Czarnkow, Poland; jszuta@steico.pl (J.S.); kszutkowski@steico.pl (K.S.)
[2] Łukasiewicz Research Network, Poznan Institute of Technology, 60-965 Poznan, Poland; magdalena.czajka@pit.lukasiewicz.gov.pl (M.C.); patrycja.kwasniewska-sip@pit.lukasiewicz.gov.pl (P.K.-S.)
* Correspondence: dfuczek@steico.pl; Tel.: +48-882116407

**Abstract:** The aim of the presented research work was to determine and analyze emissions of volatile organic compounds (VOCs) from experimental lightweight wood fiber insulation board produced in dry technology. Until now, there have been no rigid insulation materials made of wood fibers produced in such low density and made in dry technology. Among the typical parameters such as thermal conductivity and the mechanical performance of the lightweight board, attention was also paid to their influence on indoor air quality. Therefore, an attempt was made to determine the kind of substances emitting from wood fiber insulation boards produced at defined production parameters as well as the dynamics of emission reduction over time. Additionally, the influence of fire retardants used for protection against lightweight wood fiberboard fires on the emission of VOCs was analyzed. Tests on VOC emissions were carried out using the chamber method according to the applicable ISO 16000 standards. The main components emitting from lightweight insulation fiberboards were acetic acid and aldehydes such as pentanal, hexanal, heptanal, octanal, nonanal, decanal, furfural, and benzaldehyde. The percentage of acetic acid in total volatile organic compounds (TVOCs) was within the limits of 17% to 65%. From the aldehydes group, the most concerning substance was furfural due to a very strict limit value. In the presented research, depending on the variant, the emission of furfural was from 0 up to 10 μg/m$^3$ after 28 days of measurement. Other substances such as terpenes or aromatic hydrocarbons were at a very low level. The reduction in VOCs over a period of 28 days was significant in most cases from 22% up to 61%. The tests carried out also showed a substantial impact of fire retardant, used in the production of lightweight insulation fiberboard, on the emission of VOCs from fiberboards, and thus on their quality.

**Keywords:** VOCs; LDFs; wood fiberboard; lightweight insulation; acetic acid; furfural

## 1. Introduction

Nowadays, most people spend almost 90% of their lives in both residential and public buildings—schools, offices, shopping centers, etc. Indoor air quality can therefore greatly affect our health. Guided by this thought, the quality of indoor air, apart from the elements of interior design, is significantly affected by the building materials used. These include, among others, insulation boards from wood fibers. Wood fiber is an excellent, ecological, and in most cases, a semi-finished product, due to its chemical composition and morphology, which are related to their specific properties, and can be the basis for many interesting applications [1]. Because of its unique properties, wood fiber is also used for the production of insulation panels in dry technology. The boards are lighter and can be produced in greater thicknesses than, for example, boards produced in wet technology. Therefore, they are often used to insulate the building envelope, e.g., for above-rafter insulation or facade insulation, which is of great importance for an energy-efficient and comfortable indoor environment as the envelope accounts for 50–60% of total heat gain/loss in a building [2].

Additionally, wood fiber insulations ensure keeping the cold out, buffering external heat, and regulating moisture content, but also help to promote breathable structures.

Thermal insulation products are not normally exposed to direct exposure to indoor air, as they are covered with various materials such as plasterboard, wood, bricks, or concrete, and potential emissions from insulation materials must not come into contact with indoor air. However, the covering layers may not be gastight or may be perforated to accommodate technical building systems. Additionally and importantly, the building owner has the right to be informed about the quality and potential hazards of the insulation and construction materials used in their building. When referring to the quality of these materials, in addition to their physical and mechanical parameters, their impact on the environment through the emission of VOCs is also often taken into account.

Reducing the emission of VOCs from building materials, and thus improving their quality, is achieved by establishing and introducing requirements that "healthy and environmentally friendly buildings should meet". In the European Community, for many years, this activity has been dealt with by specially appointed commissions, agencies, and organizations. Since 1991, the European Collaborative Action (ECA) has been issuing reports under the common name "Indoor Air Quality and Its Impact on Man", which define and consider all aspects of the indoor environment, including: thermal comfort, sources of pollution, quality and quantity of chemical and biological indoor pollutants, energy consumption, and ventilation processes that may interact with indoor air quality [3]. The World Health Organization (WHO) [4,5] is also actively involved in the work to improve the quality of indoor air, supported by scientists from around the world, who, based on many years of research, have established among others permissible levels of concentrations of individual chemical substances, e.g., formaldehyde, VOCs in enclosed rooms, and their impact on the comfort of people within them [6–8]. Formaldehyde is classified as a carcinogen. As a result of long-term exposure, it has a negative effect on the respiratory and nervous systems of humans. Emitted from interior furnishings, e.g., furniture, it can cause severe allergies, headaches, and shortness of breath for users [9–11]. In 2016, the WHO began work on updating the global guidelines for indoor air quality, supported by numerous international organizations, e.g., the European Commission (DG-Environment), the German Federal Ministry for the Environment, Nature Conservation, Building and Nuclear Safety, the Swiss Federal Office for the Environment, and the U.S. Environmental Protection Agency (EPA) [12].

The effect of these and many other activities was the introduction of regulations allowing for the assessment of the performance of construction products and their admission to the market in the EU Member States. They are described in detail in Regulation 305/2011 of the European Parliament and of the EU Council, which "lays down harmonized conditions for the marketing of construction products" [13]. According to this document, construction products must not pose a threat to the health and safety of users, including in terms of the emission of hazardous substances (including VOCs).

In some EU countries, requirements have been introduced, specifying the permissible levels of VOC emissions from building materials and, consequently, labeling these products as ecological. For example, in Germany, the criteria for assessing VOC emissions from building materials are set by the Committee for the Health Assessment of Building Materials (Auschuss zur gesundheitlichen Bewertung von Bauprodukten-AgBB) [14]. In France, since 2011, building materials and interior furnishings (e.g., floors, windows, doors) are marked according to four classes (C,B,A,A+) depending on the permissible concentration levels specified for ten VOCs.

VOC emissions from building materials, including wood products, occur both in the phase of their use and during the processing of wood. As a result of elevated temperatures, e.g., during the drying of wood or in the process of board-pressing aldehydes, alcohols and organic acids are released [15,16]. Therefore, emissions from products and processes are distinguished [17–20].

In the case of wood-based panels, VOC emissions consist primarily of terpenes, aldehydes, and ketones, as well as organic acids. Quantitatively, the most significant monoterpenes of coniferous wood are: α- and β-pinene, limonene, 3-carene, myrcene, and β-phellandrene.

The source of terpenes is primarily the resin present in the wood of coniferous species—the raw material most often used for the production of boards [21]. In turn, aldehydes are formed as a product of the autoxidative cleavage of free unsaturated fats and fatty acids during the hydrothermal treatment of wood [22]. The unsaturated fatty acids present in the wood of coniferous species, i.e., oleic, linoleic, and linolenic, are the source of aldehydes such as octanal and nonanal, hexanal, 2-octenal, and 2-heptenal, among which hexanal dominates quantitatively [23,24]. VOC emission from coniferous wood is also affected by a part of the cross-section of the trunk from which the sample is taken. Due to the diverse chemical composition of the heartwood and sapwood, and the different anatomical structure, the compounds emitted from these zones differ qualitatively and quantitatively [25–28]. For example, terpenes are mainly emitted from the heartwood of pine. Quantitatively, there are several to several times more of them compared to their emissions from sapwood [29,30].

As mentioned earlier, all wood-based panel production processes accompanied by an increase in temperature intensify the emissions of individual organic compounds. An example is acetic acid, which is formed by the hydrolysis of the acetyl groups of hemicelluloses during the drying of wood. The hydrolysis can be acid, alkali, or thermally catalyzed. Some authors claim that the release of acetic acid increases with increasing temperature [31]. Another compound formed under the influence of temperatures is furfural. This compound is formed from wood xylose and the reaction is accelerated by temperature [32]. On the other hand, Schulz and Lukowsky [33] found that after the thermal treatment of boards, the share of terpenes in the emission decreases, but at the same time, the emission of acetic acid and furfural increases. While Scots pine (*Pinus sylvestris* Linnaeus, 1753) wood does not emit furfural, depending on the temperatures used in technological processes, emissions of this compound from wood-based panels produced from Scots pine were determined.

Emissions of VOCs from wood building materials should be eliminated from the level of raw material selection, i.e., wood, glue, hydrophobing agents, flame retardants. In turn, minimizing the degradation of wood is possible through the selection of appropriate production parameters. In the case of low-density fiberboards produced in the dry technology, this applies to the following production stages: wood defibration, the drying of wood fibers, and board pressing.

The conducted tests of VOC emissions of wood insulation boards in environmental chambers were aimed at determining the volume of emissions, the type of emitted organic compounds, and in indirectly assessing the impact of emissions on indoor air quality. The use of appropriate components in the production of the tested new generation of insulation boards would allow them to be classified as low-emission materials, i.e., safe and ecological.

## 2. Materials and Methods

### 2.1. Lightweight Insulation Fiberboards Preparation

Lightweight insulation fiberboards used for VOC investigation were produced on a full-scale production line for low density fiberboards (LDFs) from pine wood. Produced boards had a thickness ranging from 60 to 240 mm and a density of around 80 kg/m$^3$ (±20 kg/m$^3$). The production of lightweight insulation fiberboards included the following stages: chipping of pine wood (Heinola Sawmill Machinery Inc., Heinola, Finland), plasticizing the wood chips in the defibrator preheater (Sunds Fibertech, Timrå, Sweden), refining of the chips (Sunds Fibertech, Timrå, Sweden), drying of the fibers (Steico, Czarnków, Poland), gluing of the fibers (Ekologika, Kozy, Poland/Steico, Czarnków, Poland), mat-forming (KHS, Ostroróg, Poland), pre-pressing (Steico, Czarnków, Poland/Metal-impex S.C., Poręba, Poland), hot pressing (Steico, Czarnków, Poland), and finished product processing such as edge cutting (MechCad, Huta, Poland), trimming to the required board dimensions (KHS, Ostroróg, Poland) (Figure 1).

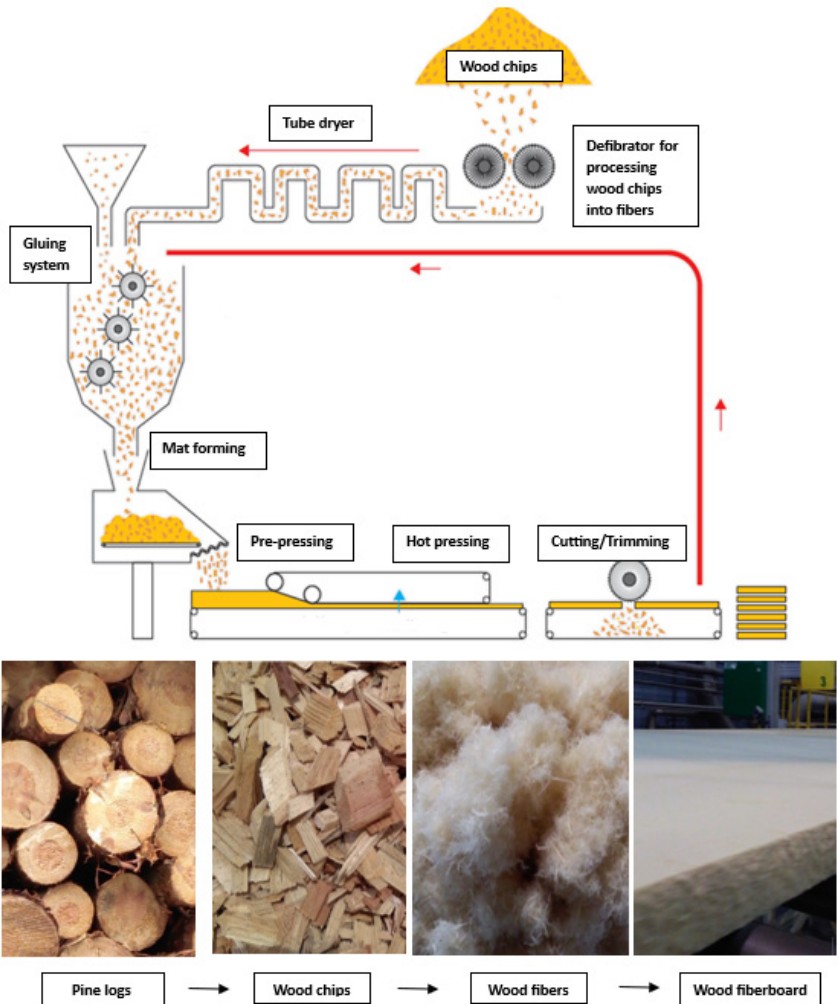

**Figure 1.** The scheme of lightweight insulation fiberboards production—dry technology.

For the lightweight insulation fiberboards production of pine wood chips, polymeric methylene diphenyl diisocyanate glue (PMDI), hydrophobization agent, and fire retardant were used. Pine wood chips used to produce the boards were conditioned to a maximum up to two weeks on the open-space raw materials area. Each time the preparation of wood fibers was carried out under the same conditions. Plasticizing of the wood chips in the defibrator preheater was conducted under a pressure of about 7 bar and temperature around 165 °C for 90 s. Afterward, the fibers were dried in a tube dryer in temperatures of 180 °C at the dryer entrance, and 80 °C at the exit. After the drying process, the fibers with a moisture content of 4.5%–6.5% were subjected to the gluing process with a PMDI-type glue in the amount of 7%. The glued wood fibers were transported to the forming station, where a wood-fiber mat was formed by the appropriate regulation of the speed of the fibers' dosing rollers to the production line, along with the proper discharge of excess fibers through the density control on the measuring system. In the next stage, an insulating mat was formed on the pre-press, which was then subjected to the process of steaming and heating in the main press with steam, with pressure parameters 6–6.5 bar and temperatures of 160–165 °C.

During the production of lightweight insulation fiberboards, three types of fire retardants (FRs) were tested:

FR 1—fire retardant based on sodium, phosphorus, and urea, pH of FR 6.5–7;

FR 2—ammonium sulfate, pH of FR 4.5;

FR 3—unknown composition, pH of FR 6.5–7.

The dosage of each FR from one hand was kept minimal to achieve the Euroclass E grade for the lightweight insulation fiberboards (rating according EN 13501-1 system) [34], and from the other hand, was kept rational from an economical and production point of view. All FRs were dosed at the same dosage point to the blow line before the tube dryer.

### 2.2. VOC Measuring Method

Test samples for the determination of VOC emission were cut in such a way that the total surface area of the cross-section planes exposed in a test chamber was 23 $m^2$ (loading factor 1 $m^2/m^3$).

All determinations of VOC emission from the tested materials were performed by the chamber method according to ISO 16000-6 [35] and EN ISO 16000-9 [36]. The test chambers with a size of 23 $m^2$ were constructed of steel (Weiss Umwelttechnik GmbH). All experiments were carried out under the following conditions: temperature of $23 \pm 1$ °C; relative humidity of $50 \pm 5\%$; air exchange rate of 0.5 $h^{-1}$; and area specific air flow rate of 0.5 $m^3/m^2 \times h$. The samples of lightweight insulation fiberboards were placed inside the chamber for the whole duration of the testing periods. A sampling of chamber air was carried out after 3 and 28 days using a portable pump, FLEC AIR PUMP 1001 (Markes International Ltd., Llantrisant, UK). The sampling flow rate was set at 100 mL $min^{-1}$. Three liters of air were collected during 30 min of sampling on sorbent tubes with Tenax TA. The glass tubes (L $\times$ R = 89 mm $\times$ 3.2 mm; Perkin Elmer, Inc. Waltham, MA, USA) were packed with Tenax TA (with 200 mg fillers; Sigma-Aldrich, Bellefonte, PA, USA). Before sampling, the tubes were conditioned at 270 °C for 15 min and exposed to the flow of helium at 50 mL $min^{-1}$. The qualitative analysis of the Tenax tubes was carried out using the gas chromatography/mass spectrometry (GC/MS) system (PerkinElmer Clarus 680, Waltham, MA, USA) equipped with a thermal desorber with cold trap injector TurboMatrix ATD (Perkin Elmer, Waltham, MA, USA). All tubes used for the sampling of chamber air and background were analyzed before experiments. The analyses were carried out under conditions given in Table 1.

**Table 1.** Analytical parameters for thermal desorber and GC-MS.

| Parameter | System |
|---|---|
| Cold trap | filled with Tenax by manufacturer |
| Tube desorption | 15 min |
| Desorb flow | 50 mL $min^{-1}$ |
| Desorption temperature | 280 °C |
| Carrier gas | He, 100 kPa |
| Column | Elite-5MS (30 m $\times$ 0.25 mm $\times$ 0.25 µm) |
| Temperature program in GC oven | 35 °C (4 min) 5 °C/min $\rightarrow$ 140 °C (0 min) 12 °C/min $\rightarrow$ 240 °C (3 min) |
| Inlet system | direct online |
| The temperature of the emission source | 220 °C |
| Electron ionization | 70 eV |
| Electron energy | TIC (scan), *m/z* |
| Scan range | 35–550 amu |
| Scanning frequency | 1 s |

Identification of the compounds was based on the database the National Institute of Standard Technology Mass Spectral (NIST MS) library. Moreover, mass spectra and retention times were compared with those of reference compounds. The concentration of selected individual VOCs was calculated based on the response factor of toluene and expressed as toluene equivalents (TEs).

### 3. Results and Discussion

Wood building materials, along with interior design elements, are an important source of emissions of harmful organic compounds into indoor air, about 60% of which make

up TVOCs [15]. In recent years, the use of ecological wood-based building materials has become increasingly popular. These include, among others, insulating materials made of wood fibers, which are an interesting alternative to the products used so far, for example, mineral wool, polystyrene, and PUR foam. Wood fiber insulation boards emit similar groups of VOCs as other wood materials. Typically, VOCs are derived from wood components and the adhesive used [37]. Other additives, especially flame retardants, which were necessary in the production of a lightweight insulation board, may also affect the decomposition of wood in certain technological conditions, causing an increase in the emission of VOCs. The main detected substances having concentrations of more than 0.1 μg/m$^3$ emitting from produced lightweight insulation wood fiberboards are listed in Table 2. The emission of most presented substances in Table 2 from insulation wood fiberboards is on a very low level compared to the lowest concentration of interest values (LCIs) given in the AgBB scheme (evaluation procedure for VOC emissions from building product). Identified VOCs have been categorized as aromatic hydrocarbons, terpenes, glycol esters, aldehydes, ketones, and acids. For further discussion, the substances which are dominantly influencing TVOCs, such as acetic acid, or are near to the limit value given by AgBB, such as furfural, are considered. Additionally, propylene carbonate was taken into account as a substance deriving from glue used for the production of the boards. Other substances due to their very low emission will be discussed in the group of compounds to which they belong.

**Table 2.** Main VOCs produced with wood fiber insulation.

| Type of Compounds/VOC Compounds | CAS No. | LCI (μg/m$^3$) |
|---|---|---|
| **Aromatic hydrocarbons** | | |
| Toluene | 108-88-3 | 2900 |
| o-cymene | 527-84-4 | 1000 |
| n-Hexane | 110-54-3 | 4300 |
| n-Heptane | 142-82-5 | 15,000 |
| **Terpenes** | | |
| 3-Carene | 498-15-7 | 1500 |
| alpha-Pinene | 80-56-8 | 2500 |
| beta-Pinene | 127-91-3 | 1400 |
| Limonene | 138-86-3 | 5000 |
| **Glycols, Glycol ethers, Glycol esters** | | |
| Propylene carbonate | 108-32-7 | 1000 |
| **Aldehydes** | | |
| Hexanal | 66-25-1 | 900 |
| Octanal | 124-13-0 | 900 |
| Nonanal | 124-19-6 | 900 |
| Decanal | 112-31-2 | 900 |
| Pentanal | 110-62-3 | 800 |
| Butanal | 123-72-8 | 650 |
| Benzaldehyde | 100-52-7 | 90 |
| Furfural | 35796 | 10 |
| **Ketones** | | |
| Acetone | 67-64-1 | 1200 |
| **Acids** | | |
| Acetic acid | 64-19-7 | 1200 |
| n-Caproic acid | 142-62-1 | 2100 |

In Table 3, the variants of VOC measurements are presented. Samples from numbers 1 to 5 were produced with the use of the same fire retardant based on sodium, phosphorus, and urea; sample number 6 was produced using ammonium sulfate as a fire retardant; and

the last sample, number 7, was produced using a fire retardant with an unknown recipe. All the samples met the requirements of the AgBB scheme regarding TVOCs limit after 28 days—1000 $\mu g/m^3$ and for individual substances based on LCI values. After 3 days, all TVOC results were within the AgBB limit of 10 000 $\mu g/m^3$, and the single VOC of furfural emitted from samples 5 and 6 exceeded the permissible limit by 10 $\mu g/m^3$.

**Table 3.** Chosen VOC results.

| Sample No. | 1 | 2 | 3 | 4 | 5 | 6 | 7 |
|---|---|---|---|---|---|---|---|
| Thickness of the board (mm) | 60 | 240 | 60 | 160 | 240 | 60 | 60 |
| Fire retardant (FR) type | FR 1 | FR 1 | FR 1 | FR 1 | FR 1 | FR 2 | FR 3 |
| VOC 3 d 28 d [$\mu g/m^3$] | | | | | | | |
| Acetic acid 3 d | 159 | 101 | 230 | 170 | 239 | 365 | 10 |
| Acetic acid 28 d | 26 | 89 | 95 | 61 | 149 | 218 | 16 |
| Furfural 3 d | 16 | 0 | 2 | 4 | 22 | 50 | 0 |
| Furfural 28 d | <1 | 0 | 4 | 0 | 10 | 8 | 0 |
| Propylene carbonate 3 d | 97 | 7 | 31 | - | 21 | - | - |
| Propylene carbonate 28 d | 22 | <1 | 30 | - | 5 | - | - |
| TVOC 3 d | 394 | 224 | 356 | 335 | 444 | 764 | 65 |
| TVOC 28 d | 155 | 175 | 182 | 142 | 223 | 361 | 72 |

The TVOC values after the third day of exposure for samples no. 1–5 ranged from 224 $\mu g/m^3$ to 444 $\mu g/m^3$, and for sample no. 6, the TVOC value was almost twice as high at 764 $\mu g/m^3$. In turn, the lowest VOC emission was found for sample no. 7 (65 $\mu g/m^3$). After 28 days of exposure of the samples in the chamber, a decrease in emission by about 50% was observed for most of the tested panel variants. As shown in Figure 2, the TVOC value decreased the least, by 22%, for sample no. 2, and in the case of sample no. 7, after 28 days, the TVOC value slightly increased by 10%.

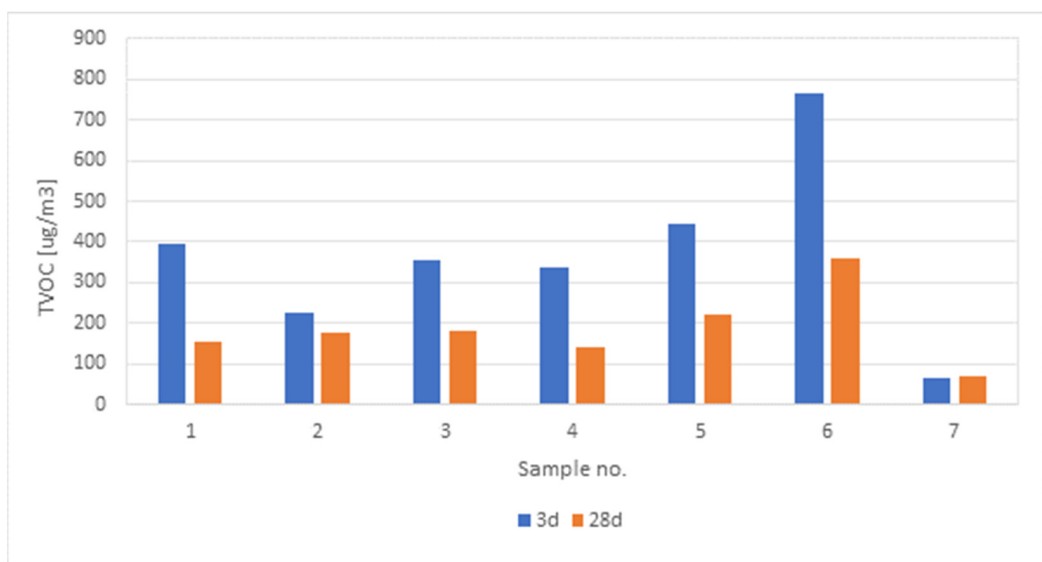

**Figure 2.** TVOC values for test samples after 3 and 28 days of exposure in the chamber.

As it is already known, the temperatures used in the technological process and the exposure duration to high temperature of the wood material—together with the raw materials used for production—are significant for the emission of VOCs [38,39]. During the production of lightweight insulation boards, the pine wood chips are converted to fibers using a hydrothermal treatment in a pressurized refiner. The temperature in the refiner is set to around 165 °C, and later in the tube dryer, rises up to 180 °C, which drive terpenes from the material, resulting in their lower emissions from the final product. This explains

the very low emission of such terpenes as α-pinene, β-pinene, 3-carene, or limonene, whose boiling points range from 155 °C to 176 °C. A similar observation was also noticed in case of medium-density fiberboards (MDFs), during which production temperatures in the pressurized refiner are maintained between 165 and 185 °C [25]. According to the literature, terpenes, under conditions of elevated temperatures, partly evaporate and partly degrade to o-cymene and p-cymene (degradation products of camphene, delta-carene, and limonene) [40]. The literature reports show that the drying of wood clearly reduces the emission of terpenes—depending on the temperature and duration of the process. Some researchers argue that the terpenes which are already released are no longer emitted by the product [32,41]. On the other hand, high temperatures in the production of lightweight insulation fiberboards can lead to the degradation of hemicelluloses and the creation of carboxylic acids (mainly acetic acid), aldehydes such as furfural, and ketones such as acetone, and others. The production of lightweight insulation boards discussed in this scientific paper was carried out under such temperature conditions to minimize VOCs from the final product and was the same for each variant.

The share of the main substances group in the TVOC for each evaluated sample is shown in Figures 3 and 4. Three days after installing the samples in the chambers, the main substance emitted from all samples, except sample no. 7, was acetic acid. The share of acetic acid in TVOCs after 3 days of measurement for samples from no. 1 to 6 was from 40% to 65%. Only for sample no. 7 was the share of acetic acid in TVOCs much lower—around 15% after 3 days and 22% after 28 days. For other samples, except sample no. 1, after 28 days, the main emitting substance was still acetic acid.

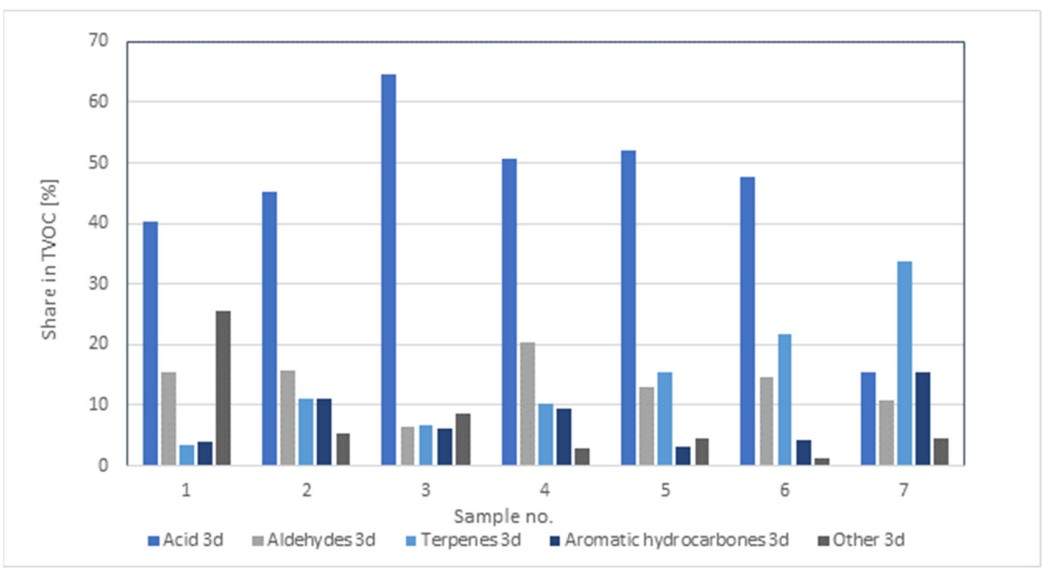

**Figure 3.** Emission of individual VOC groups from test samples after 3 days of exposure in the chamber.

The second dominant group in TVOCs were aldehydes, mainly hexanal, octanal, benzaldehyde, nonanal, and furfural. They accounted for 10 to 20% of TVOCs after 3 and 28 days of sample exposure in the chamber. A characteristic feature of aldehydes is their easily perceptible smell, even at low concentrations. According to the literature reports, hexanal, heptanal, and octanal have an extremely low odor threshold of 0.00028, 0.00018, and 0.00001 ppm, respectively. These odors, which cannot even be detected by GC-MS analysis, are perceptible to humans [22]. The only aldehyde whose concentration in the chamber air after day 3 exceeded the permissible limit by 12 μg/m$^3$ for sample 5, and by 40 μg/m$^3$ for sample 6, was furfural. After 28 days, the emission of this compound in both cases decreased to the permissible level of 10 μg/m$^3$. For the remaining test variants, furfural was not detected in the chamber air, or otherwise, its emission was at a low level (Table 3).

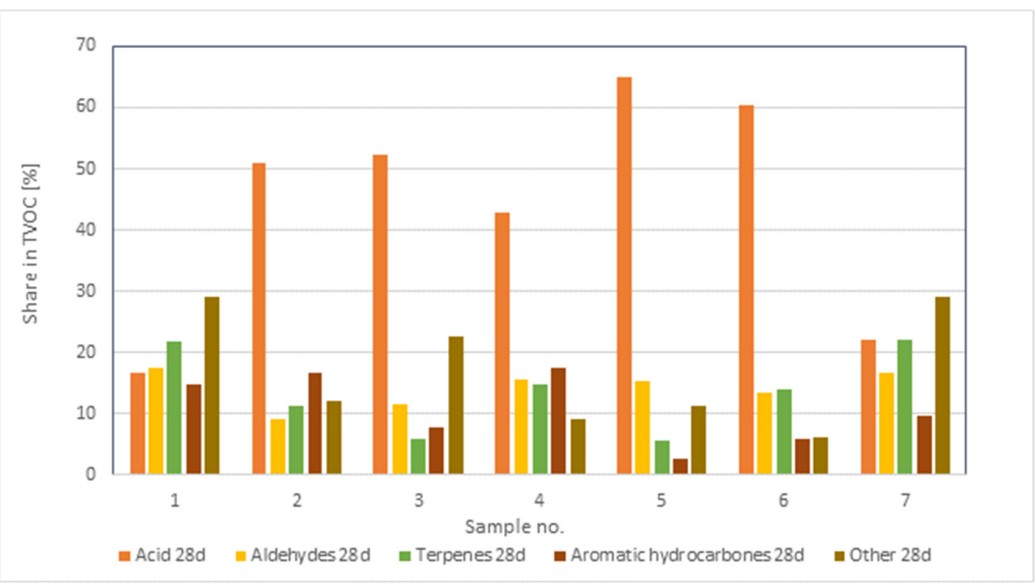

**Figure 4.** Emission of individual VOC groups from test samples after 28 days of exposure in the chamber.

For the production of wood fiber insulation boards, PMDI glue was used, which is a source of propylene carbonate emissions. It was the only compound not derived from the raw wood material used. The compound was emitted from four variants of trials, among which, the highest emission of this compound was characterized by sample no. 1 at about 100 μg/m$^3$, and a three-times lower emission was by sample no. 3 (after 3 days).

During the production of lightweight wood fiber insulation boards, the influence of the different types of fire retardants on VOC emission was tested. The predominating influence of the tested fire retardants on VOC emission, especially on the emission of acetic acid and furfural, has an FR no. 2: ammonium sulfate. The lowest fire retardant was sample no. 3, with an unknown composition. Ammonium sulfate, as a salt of strong acid (H$_2$SO$_4$) and weak base (NH$_4$OH), creates acid conditions. When added to the blow line before drying the wood fibers, it causes additional wood degradation enhanced by an increased temperature and, additionally, an acid reaction of the fire retardant. It is beneficial to use fire retardants with a neutral reaction, which does not increase the degradation of wood at elevated temperatures and, at the same time, does not significantly affect the emission of VOCs.

Additionally, the influence of the thickness of the samples produced on the same fire retardant and on the same production parameters on VOC emission was evaluated (sample no. 1 to 5). According to the literature, there is an influence of the thickness of the wood-based panels on VOC emission even for the lower range of thickness, such as 8 to 16 mm [33]. The authors state that in the case of particleboards, the VOC emission also increases with the increase in thickness on each day of the VOC measurement. This increase in VOCs is explained by the increasing utilization of adhesive and wood particles for thicker boards. In the case of the conducted research, this dependency presented in Figure 5 was not confirmed despite the large differences in the thickness of the tested boards most likely because the production of the boards was on a real production scale (minimum capacity of the production line was 3 tons per hour), and the variety of wood material could have influenced the results.

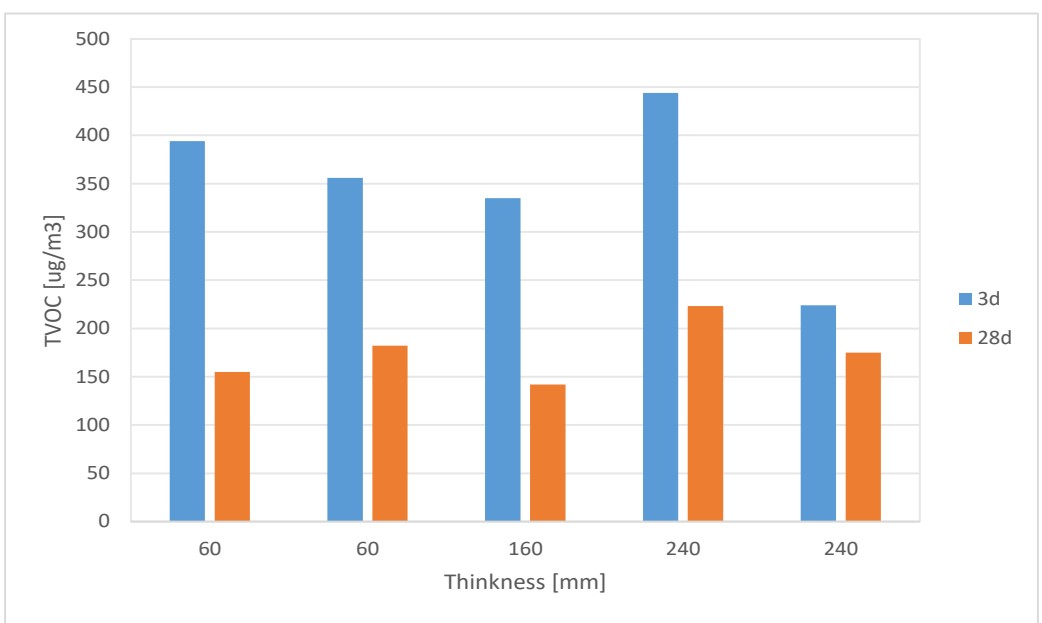

**Figure 5.** The influence of thickness of the board on TVOCs from test samples after 3 and 28 days of exposure in the chamber.

## 4. Conclusions

- VOC emission from lightweight insulation wood fiberboard is at a very low level and meets the requirements of the AgBB scheme regarding TVOC limit after 28 days—1000 μg/m$^3$ and for individual substances based on LCI values (lowest concentration of interest).
- After 28 days of exposure of the samples in the chamber, the reduction in VOCs was significant in most cases, from 22% up to 61%.
- The largest share in TVOCs for all the tested boards (except for sample no. 6) had acetic acid, although its emission was much lower than the limit of 1200 μg/m$^3$ given by AgBB.
- The influence of an acid fire retardant—ammonium sulfate—on the increase in acetic acid and furfural emission was noticeable.
- Although according to the literature there is an influence of the thickness of the wood-based panels on VOC emission, in the presented results, there were no clear dependence confirming this fact.

The practical effect of the research was the emission quality assessment of the experimental lightweight wood fiber insulation boards in terms of the requirements enforced in the European Union countries. The proposed research is part of the current global work on improving air quality (indoor air quality), which aims to reduce VOC emissions from building materials and home furnishings. This is also linked to the guidelines of the EU directives that define the conditions for placing construction products on the market. Low-emission and ecological building materials made of raw materials of natural origin are and will be willingly used in the future by both the construction industry and potential users. Currently, nearly every new material introduced to the construction market must undergo a number of tests confirming its emission quality. Dry-formed boards made of wood fibers, in addition to excellent insulating properties, have also proven to be a product that does not have a harmful effect on human health. The results presented in the article for several variants of plates, not exceeding the permissible concentrations for individual VOCs (after 28 days), are an important argument for introducing a new product to the market.

In order to better understand how VOC emission is changing over time in different storage conditions or in real conditions as an insulation of houses, it would be interesting

to continue research in this area, especially as there are not many published works in the field of insulation material from wood fibers.

**Author Contributions:** Conceptualization, D.F., M.C. and J.S.; methodology, D.F. and M.C.; formal analysis, D.F. and M.C.; investigation, D.F., M.C., K.S., J.S. and P.K.-S.; data curation, D.F. and M.C.; writing—original draft preparation, D.F. and M.C.; writing—review and editing, D.F., M.C., J.S., K.S. and P.K.-S.; supervision, D.F., M.C., J.S., K.S. and P.K.-S. All authors have read and agreed to the published version of the manuscript.

**Funding:** The research was carried out under development project (No. POIR.01.02.00-00-0099/17-00), co-funding within sectoral program WoodInn implemented by the National Center for Research and Development (NCRD) within the Operational Program Smart Growth 2014–2020.

**Data Availability Statement:** The data presented in this study are available on request from the corresponding author.

**Conflicts of Interest:** The authors declare no conflict of interest.

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
