# Peer review of "VOC Emission from Lightweight Wood Fiber Insulation Board"

_forests, doi:10.3390/f14071300_

Round 1

Reviewer 1 Report

The manuscript is focused on the investigation and evaluation of the emission of volatile organic compounds (VOCs) from insulation fiberboards, and the effect of fire retardant on the emission of VOCs. In this respect, the manuscript is within the scope of the Forests journal. In general, the manuscript is well-written, structured, and informative, but needs some minor improvements. Please, see below my comments on your work:

In general, the title (line 2), the abstract (lines 8 to 20) and the keywords (line 21) correspond to the scope, aims and objectives of the manuscript. The abstract is well-written and informative, and clearly presents the aim of the research. However, I’d recommend further extending it by providing more specific results obtained from the experimental work.

Please add numbers to the sections of your manuscript. Please refer to the Instructions for authors and template of the journal.

Line 11: fiberboard is one word, please revise throughout the manuscript.

Line 15: please provide the full term, i.e. total volatile organic compounds, then followed by the common abbreviation (TVOC).

Line 38: the correct term is volatile organic compounds (VOC), not substances.

Lines 53-57: this statement should be supported by a relevant reference.

Line 57: please define EPA. U.S. Environmental Protection Agency?

Lines 60-61: please add the cited regulation in the references of your work.

Writing about building materials and wood-based composites and their effect on indoor air quality, a few lines about the negative health effects (both in short and long term) of free formaldehyde emitted from wood composites should be added, supported by relevant references.

Line 103: please provide the botanical name of Scots pine in Italics. Why do you mention this exact species? Please explain.

Line 106: please use the already introduced abbreviation VOC.

Overall, the Introduction part is well-written, informative, and provides relevant information and references on the research topic. The aim of the research has also been clearly outlined.

In line 119 you used “fiberboards”, and in line 120 “fibreboards”: please unify the terms used.

Lines 123-127: please add relevant information about the equipment used, e.g. company producer, city, country.

Line 151: please add the standard EN 13501-1 in the references of your work.

Overall, the Materials and Methods section is well-written and detailed, providing relevant information on the materials and equipment used to perform the experiments.

Line 186: the section should be renamed to Results and discussion.

Line 230: although well-known, please add the full term, i.e. medium-density fiberboards, then followed by the common abbreviation MDF.

Line 268, Figure 4: please check and correct the caption of the figure, now it includes only 28 days of exposure, but you have also given the results obtained after 3 days.

Overall, the results are clearly presented, but their discussion with relevant research works should be extended.

The Conclusion part reflects the main findings of the manuscript. Please add the potential for future studies in the field, as well as some information on the practical application of your results.  

The References cited are appropriate to the topic of the manuscript. However, they are not formatted in accordance with the Instructions for Authors of the Journal. Please add the standards in the general list of references used, and do not provide them separately.

English language and style used are fine with only some minor issues to be addressed.

Reviewer 2 Report

In this article, aiming the VOC issues in wood fiber board field, authors investigated the VOC emission from lightweight wood fiber insulation board. Comprehensive characterizations and analysis have been performed. In general, the manuscript is well prepared and the conclusion is supported by the experimental and results. However, there are still some issues to be addressed.

1.         One or two sentences are required at the beginning of abstract to present the background or aim of this work.

2.         It is suggested to add more keywords.

3.         Three-line tables should be applied for a better scientific expression.

4.         One scheme to show the experimental procedure is suggested for better understanding of this work to readers.

5.         More background on structure, properties, and applications the wood fiber board should be provided: HAO Xiaolong, ZHOU Haiyang, SUN Lichao, LIN Dongrong, OU Rongxian, WANG Qingwen. Research progress and application of co-extruded wood plastic composites[J].Journal of Forestry Engineering,2021,6(05):27-38.doi:10.13360/j.issn.2096-1359.202011003; Molded fiber and pulp products as green and sustainable alternatives to plastics: A mini review; JIA Shifang, LIU Jingyi, LIN Xianxian, SUN Weisheng, GUO Xi, CAO Huimin, WANG Wenbin. Sound absorption performance of bionic perforated wood structure fiberboard[J].Journal of Forestry Engineering,2021,6(01):38-43.doi:10.13360/j.issn.2096-1359.202004022; etc.

6.         There are too many too old references, which is better to be deleted or replaced with recent articles to show the novelty of this work.

7.         There are still some typos and grammar issues in the manuscript, especially the use of blank space. Authors should carefully recheck the whole manuscript.

8.         Some figures should be modified with addition of error bars.

9.         It is suggested to write the conclusion section in one paragraph in a fluent logic way.

10.     Authors should recheck the references to make sure full information is provided, such as volume, pages, etc. In addition, the format of references should be uniform.

Round 2

Reviewer 2 Report

Accept in present form